# Conceptualizing Epigenetics and the Environmental Landscape of Autism Spectrum Disorders

**DOI:** 10.3390/genes14091734

**Published:** 2023-08-30

**Authors:** German Torres, Mervat Mourad, Saba Iqbal, Emmanuel Moses-Fynn, Ashani Pandita, Shriya S. Siddhartha, Riya A. Sood, Kavya Srinivasan, Riya T. Subbaiah, Alisha Tiwari, Joerg R. Leheste

**Affiliations:** 1Department of Counseling and Clinical Psychology, Medaille College, Buffalo, NY 14214, USA; gthominid5@gmail.com; 2Department of Clinical Specialties, New York College of Osteopathic Medicine, Old Westbury, NY 11568, USA; mmourad@nyit.edu; 3Department of Biomedical Sciences, New York College of Osteopathic Medicine, Old Westbury, NY 11568, USA; siqbal10@nyit.edu (S.I.); emosesfy@nyit.edu (E.M.-F.); apandita@nyit.edu (A.P.); rsood01@nyit.edu (R.A.S.); ksrini01@nyit.edu (K.S.); atiwar06@nyit.edu (A.T.); 4Dedman College of Humanities and Sciences, Southern Methodist University, Dallas, TX 75275, USA; shriyasiddhartha@gmail.com; 5Department of Arts and Sciences, Georgetown University, Washington, DC 20057, USA; riyasubbaiah17@gmail.com

**Keywords:** alleles, brain, disease, evolution, genes, gastrointestinal tract, mutations, phenotypes, epigenetics

## Abstract

Complex interactions between gene variants and environmental risk factors underlie the pathophysiological pathways in major psychiatric disorders. Autism Spectrum Disorder is a neuropsychiatric condition in which susceptible alleles along with epigenetic states contribute to the mutational landscape of the ailing brain. The present work reviews recent evolutionary, molecular, and epigenetic mechanisms potentially linked to the etiology of autism. First, we present a clinical vignette to describe clusters of maladaptive behaviors frequently diagnosed in autistic patients. Next, we microdissect brain regions pertinent to the nosology of autism, as well as cell networks from the bilateral body plan. Lastly, we catalog a number of pathogenic environments associated with disease risk factors. This set of perspectives provides emerging insights into the dynamic interplay between epigenetic and environmental variation in the development of Autism Spectrum Disorders.

## 1. Clinical Vignette Case

A 3-year-old white male is brought in by his parents to the pediatrician for multiple behavioral concerns that have gradually developed in the past few months. According to his parents, the child shows signs of cognitive impairment, including poor social interactions with family members, failure to understand social cues, struggles with starting and maintaining conversations, and limited vocabulary skills for his age. In addition, the child is hypersensitive to sounds and light, which is quite upsetting and overwhelming. The most disturbing behavioral concern, according to his parents, is the recurrent body movements, such as hand-flapping and rocking back and forth. Other obtrusive behavioral symptoms include intense focus on specific objects, emotional outbursts of agitation and frustration, and overall resistance to change. The parents also report that their son has had difficulty reaching age-appropriate milestones such as walking and coordinating fine and gross motor skills. The patient has been noted to hold his bowel movements for several days and reports significant abdominal pain when stooling. Family history is negative for any neurological or psychiatric disease. Routine laboratory workups are unremarkable. Parents report that the child is the second sibling; an older 6-year-old sister is described as being age-appropriate in terms of overall cognitive function and emotional development. The mother reports that her pregnancy with the patient was particularly stressful due to the emotional strain of two successive miscarriages. To ensure an uneventful pregnancy, the mother had to undergo hormonal therapy (human chorionic gonadotropin, HCG) and bed rest. The child was born full term via Cesarean section due to breech presentation.

## 2. Introduction

Developmental neuropsychiatric disorders affect structural and functional connectivity in several cell networks known to play a significant role in cognitive performance (e.g., cortical neurons) and emotional processing (e.g., amygdaloid neurons); pathological dysfunctions are used as diagnostic and typing criteria for the clinical management of the disorder. Although the specific origins of early-life disorders are largely unknown, they are thought to nevertheless occur directly via genetic mutations that have disease-driving effects, irreversible developmental deviations, or brain injuries that lead to premature cell death or necrosis. Developmental neuropsychiatric disorders can also occur indirectly through epigenetic changes that mark DNA strands in future generations for heightened disease susceptibility, especially at juvenile stages. In addition, aberrant cell-to-cell (e.g., ovum-sperm) signaling events, either by direct contact with each other or by the secretion of downstream molecules, such as integrin glycoproteins, may account for some aspects of neuropsychiatric symptoms in pre-pubescent offspring. Regardless of their origin, developmental psychiatric disorders comprise a number of debilitating diseases with varied clinical and behavioral outcomes [1]. Although the brain is thought to be the primary source for disease onset, it is likely that the bilateral body plan (i.e., the left side is a mirror image of the right side) is also critically involved in the progression and manifestation of aberrant or maladaptive behavior.

Although gene variants that influence disease risk for developmental neuropsychiatry disorders appear to have hominin-specific origins, the mammalian brain and bilateral body plan networks that they influence have ancient evolutionary histories. Indeed, cell networks such as those in the amygdala and gastrointestinal tract can be traced back to evolutionary time scales when early hominin lineages had not yet appeared [2,3]. Thus, it is conceivable that gene variants that carry disease-driven effects have only recently become coupled with the ecological, cultural, and/or demographic histories of modern human populations. If that is the case, then Autism Spectrum Disorders (ASDs), for example, may be viewed as diseases that have sensitive genes whose expressions are strongly associated with particular environmental conditions. Because ASD produces symptoms of emotional, perceptual, and cognitive dysfunction, it is likely that ASD first emerged in naive populations (i.e., populations without the disorder) with large and complex social formations. Such phenotypic coupling implies that specific combinations of genes and environmental risk factors can increase the chance or likelihood of developing a particular disease whose behavioral dysfunctions become prominent prior to sexual maturity.

Considerable research progress has been made in exploring the underlying genetic mechanisms of ASD. However, many preclinical and clinical studies are still needed to gauge how gene variants contribute specifically to a particular maladaptive behavior. This knowledge is crucial for understanding the pathological changes that contribute to disease onset, disease progression, and ultimately, the application of individualized gene-targeting therapy. As described above, ASD may be the result of hominin-specific gene variants and modern social contexts. This ‘gene by social interaction’ can be best understood through the lenses of epigenetics, as environmental factors can actually affect whether and how genes are transcribed in mammalian cell networks [4,5]. Indeed, injurious experiences during human development can lead to structural and chemical changes in the brain and bilateral body plan through epigenetic modifications that can be maintained from cell to cell and, in some cases, can be inherited across generations [6]. Note: Bibliographic searches on PubMed, Google Scholar, and Scopus were performed, and publication dates were limited to 2015–2023, except for some benchmark publications. Literature searches were performed using keywords pertaining to the scientific foundation and clinical diagnosis of ASD. A total of 84 articles were retrieved and included in this conceptualizing perspective.

## 3. Epigenetics: An Overview

Human DNA, like any other placental mammalian genome, has been shaped by Darwinian evolution in a lineage-specific manner. Thus, mammalian genomes share many protein-coding sequences, including their start and stop codons. However, many other DNA variants are more constrained to human genomes, including transcription factor binding sites and non-coding regulatory elements. Thus, long non-coding RNA promoters and exons are preferentially conserved in human genomes, which may explain either the recent evolution of disease-causing DNA variants or the inherited liability of developmental neuropsychiatric disorders such as ASD [7]. More recently, however, the identification of epigenetics in disease has allowed us to envision this form of gene transcription reaching the fetal stage and impacting the evolution of a juvenile’s susceptibility to brain ailments. Epigenetics refers to changes in DNA that do not involve modifications to the underlying nucleotide base sequence of a particular gene. Importantly, these changes are reversible and highly susceptible to evolutionary selection pressures. Thus, epigenetic mechanisms regulate either the ON or OFF expression of genes according to prevalent environmental and/or behavioral conditions (Figure 1). In this context, three epigenetic reprogramming signatures have been identified in eukaryotic cells: DNA methylation, histone modification (acetylation/deacetylation), and non-coding RNA activity. To illustrate this reprogramming concept, high DNA methylation levels, particularly at the promoter region, and low histone acetylation levels reduce transcription rates by diminishing the accessibility of DNA protein-coding sequences (OFF conformation) [7,8,9]. Conversely, low levels of DNA methylation at the promoter region and high levels of histone acetylation increase the probability of gene transcription (ON conformation). Methylation is the process by which methyl groups (CH3) are enzymatically added to specific locations within the DNA chemical structure. On the other hand, histone deacetylases are nuclear enzymes that eliminate acetyl groups (-COCH3) from both histone and non-histone proteins in response to various intracellular and extracellular signaling events. In general, both of these chemical pathways regulate the ON and OFF synthesis of RNA species and functional proteins that genes encode. Against this background, it is thought that the dysregulation of the aforementioned epigenetic signatures, in particular DNA methylation and histone deacetylation, may be linked to the pathophysiology of ASD [7,8,9].

## 4. DNA Methylation and Histone Deacetylation in ASD

During brain development, DNA methylation in neurons oscillates between hypermethylation and hypomethylation states of physiological activity. As a consequence, differentiated neurons develop a stable and unique DNA methylation signature that guides genomic imprinting, transcriptional regulation, and chromosome stability (Figure 2) [10,11,12]. The cytosine-phospho-guanine (CpG) dinucleotide sequence of the PPP2R2C gene is hypermethylated, according to analysis of DNA methylation across the autism genome [4,5,6]. Conversely, hypomethylation of the CpG dinucleotide sequence is also associated with ASD, as loss of binding specificity to this particular genomic sequence leads to dysregulation of nuclear and cytoplasmic genes involved in metabolic and mitochondrial activity, respectively [5]. Thus, there is a critical balance between variable epigenetic states that is established early in development and maintained through subsequent cell divisions [10,11,12]. If the physiological states of DNA methylation are altered by environmental risk factors, especially during gestation, neuron type-specific DNA hypermethylation or hypomethylation will occur, leading to sustained changes in gene expression and function [11,12,13].

Removal of -COCH3 groups from specific amino acid residues is an additional mechanism of epigenetic marking. Such removal causes the DNA helix around histone proteins (i.e., the chromatin structure) to become more tightly packed and therefore less accessible to gene transcription. Evidence, primarily from animal and in vitro studies, suggests that deacetylation of histone 3 proteins at lysine residues (h3k27ac) results in abnormal transcription of candidate genes implicated in ASD [14,15]. Interestingly, histone deacetylase inhibitors such as Valproic Acid, a drug used primarily in the treatment of epilepsy and bipolar disorder, can increase the risk of congenital malformations in infants exposed to the antiepileptic drug in utero [16,17,18]. This effect may be caused by the widespread enhanced accessibility of the chromatin structure to genomic DNA [16,17,18]. Thus, chromatin accessibility to regulatory signals is a source of epigenetics. In general, these findings reveal how chromatin accessibility to both intracellular and extracellular cues can regulate epigenetic function and drive abnormal cell morphology and dysfunction in disease.

## 5. Homeodomain-Based Strategies for the Identification of Genes Associated with ASD

Homeobox genes are a large and highly specialized group of closely related genes that drive the development and formation of the bilateral body plan, including the brain [19,20,21]. Homeobox genes encode homeodomain transcription factors that broadly regulate gene expression in cell clusters, tissue arrays, and organ systems. In this context, there is evidence that certain homeobox genes might be linked to the onset of ASD. To expand this evidence further, we list here fourteen homeodomain-containing genes using GeneCards’ Inferred Functionality Scores (GIFtSs) for the identification of novel autism-associated genes. As shown below, the assigned genes have functional scores of 50 or higher, indicating a significant amount of functional knowledge. Of interest, these homeodomain-containing genes (Table 1) are known to be functionally involved in embryonic brain development [19,20,21].

Except for the DLX5 homeobox gene, all identified genes interact with the transcription factor SP1 (Genome Browser, University of California, Santa Cruz) [22,23,24,25,26,27,28,29,30,31,32,33,34,35]. SP1 is known to regulate several autism candidate genes [36,37,38,39,40,41], and its biological importance to ASD is underscored by the fact that SP1 proteins are markedly elevated in the anterior cingulate gyrus (ACG) of autistic patients [42]. This cortical brain region has specialized subdivisions of function that extend far into the neural territories of the amygdala and orbitofrontal cortex. As discussed below, the amygdaloid complex and orbitofrontal cortex are brain networks underlying the nosology of ASD [42]. Functionally, the ACG directs emotional and cognitive activity in social environments, and therefore, abnormalities in the structural and/or chemical composition of ACG neurons may lead to inappropriate emotional behaviors. In general, these data illustrate the genetic component and polygenicity underlying most developmental neuropsychiatric disorders.

## 6. Clinical Case Presentation of ASD

Currently, there are hundreds of gene variants associated with the onset of ASD. Thus, the clinical presentation of the disease is highly variable, as one or more behavioral symptoms may be related to other comorbid conditions (e.g., anxiety disorder, seizure disorder) besides autism. In addition, antagonistic pleiotropy and dosage-sensitive genes further fragment the phenotypic characteristics of ASD. Regardless, here, we present a prototypical autism clinical vignette with five behavioral specifiers: cognitive disability; deficits in social–emotional reciprocity; repetitive or stereotyped motor behavior; improper coordinated language communication; and gastrointestinal distress. Underneath this clinical vignette, we microdissected and correlated a particular phenotype of the disease to functionally and anatomically related regions of the brain and bilateral body plan. The structural organization imposed here will not only identify a wide network of cells, but also specific clusters of genes targeting a particular symptom within behaviorally relevant regions. It is expected that such structural organization will help lay a solid foundation in psychiatry and point to more focused approaches to a deeper understanding of ASD and its individualized treatment (Table 2).

Based on the clinical vignette just described, we microdissected the following brain regions pertinent to the nosology of autism: the lateral and orbital frontoparietal, insula and cingulate gyrus regions of the cortex. These cortical subregions appear to be involved in the fundamental features of cognitive function [46]. The amygdaloid complex within the mid-temporal lobe is a subcortical region that is critical for emotional processing and motivated behavior [15]. The basal ganglia (caudate nucleus, putamen, sub-thalamic nucleus, globus pallidus, and substantia nigra reticulata) is a network of neurons that connect directly to specialized cortical cells, thus processing higher-order motor–cognitive function [14]. The frontostriatal and/or frontocerebellar circuits are an extensive network of interconnected neurons underlying speech and language production for both lexical and semantic categories [47,48,49]. Smooth muscle cells, enteric neurons, interstitial cells of Cajal and telocytes of the gastrointestinal tract control the musculature and transmucosal fluid movement of the tract, thus coordinating digestive and motility activity [50,51,52] (Figure 3).

## 7. Genomic Imprinting in ASD

The brain regions listed above show a number of epigenetic states that are brain-region-, cell type-, and age-dependent. Indeed, some of these epigenetic states have been previously associated with developmental disorders, and some of them have been linked directly to ASD. By-and-large, the identified epigenetic states are functionally involved in cell development, cell architecture, cell fate, and, importantly, disease risk. Because epigenetic imprinting involves the monoallelic inheritance of maternal or paternal genes, the emerging picture is that some subtypes of ASD are maternally derived from the duplication of imprinted domains on chromosome 15q11–13 [53]. Another monoallelic inherited gene thought to be a risk factor for ASD is Autism Susceptibility Candidate 2 (*AUTS2*), which is also duplicated in lymphoblastoid cell lines derived from autistic patients [54]. In this context, the duplication and deletion of DNA sequences (collectively referred to as copy number variants, or CNV) are thought to underlie a number of neuropsychiatric disorders as CNV modulates levels of disease penetrance and expressivity among affected individuals. Thus, based upon these findings, it is sensible to ask whether allele-specific expression and epigenetic states are sex-biased, with females transmitting a heavier mutational load than males. Although this assumption has become more apparent through the use of comparative genomics, confirmation of this hypothesis remains elusive [55,56]. Regardless, monoallelic inheritance and epigenetic imprinting sculpt the autistic brain to be a mosaic of neurons, with some harboring wild-type alleles while others harbor mutant alleles, where both sets of alleles produce different proteins through alternate splicing in the cortex, amygdala, basal ganglia, and gastrointestinal tract. In the broader context of clinical implications, it is conceivable that imprinted alleles are particularly more susceptible to early environmental insults and disease than unimprinted (i.e., wild-type) genes. Concomitantly, it is also feasible that monoallelic genes accumulate higher DNA methylation levels compared to hypomethylated biallelic genes.

## 8. Sex-Dependent Differences in ASD

Other quantitative observations from the spectrum of autism are the overall incidence of the disease, males being more likely than females to inherit the disease, and the risk of recurrence for the disease to affect closely related cohorts. As of 2020, the CDC reports that 1 in 36 children in the USA are diagnosed with ASD [57,58]. Meticulous screening practices, broader diagnostic criteria, and the inclusion of minority patients in metric pools appear to account for the increased incidence of autism [3,4]. The disease is estimated to be four times more common in males than females. This skewed distribution could partially be attributed to the unequal inheritance of mitochondrial DNA (mDNA) and the gut microbiome [59,60]. In both instances, mDNA and communities of bacteria and viruses are inherited only from mothers. This unusual mode of inheritance makes male offspring more susceptible to harmful mutations and dysbiosis, respectively. Other key differences between males and females are genes found in male chromosomes, such as *SOX9,* which is a direct target of *SRY*. *SRY* is the mammalian Y-chromosomal testis-determining gene that guides male sex determination [61]. Moreover, androgen hormones (e.g., testosterone) play a key role in the development and maturation of the male brain, the bilateral body plan, and sexual motivation and behavior [16,62]. All of these sex-dependent differences are important for assessing CNV risks, epigenetics, and gene-by-environment interactions in ASD. Finally, there is a growing awareness that exposure to pathogenic environments such as smoking, drugs, and nutrition affects the phenotype of subsequent generations through the reprogramming of the epigenetic signature of spermatozoa. Indeed, there is evidence of high DNA methylation levels in the sperm cells of fathers who have already had one child diagnosed with autism [16,63]. The genes targeted for methylation are mostly DNA variants involved in embryogenesis, in particular brain ontogenesis. Thus, the preconception period is a sensitive developmental window for the reproductive health of fathers. If chemical insults or untoward lifestyles occur during the course of preconception periods, epigenetic modifications will gradually accumulate in haploid sperm cells. Eventually, this altered methylated legacy will determine the risk of ASD in future generations.

## 9. Assisted Reproductive Technology and ASD Risks

Epidemiologic studies focused on nongenetic factors have established advanced parental age, maternal health, Cesarean delivery, preterm birth, and fetal distress as ASD risk factors [7,64]. Of potential interest, some studies suggest an association between autism risk and assisted reproductive technology. For example, in vitro fertilization, intracytoplasmic sperm injections, zygote intrafallopian transfer, gamete intrafallopian transfer, and artificial insemination may activate dormant molecular, often epigenetic, processes within growing oocytes, leading to measurable short-and long-term risks for the unborn child [64,65]. Obviously, the same risks for inducing epigenetic and imprinting disturbances in vitro apply to spermatozoa, as these germ cells undergo similar but not identical culture procedures as oocytes. In general, obstetrical prenatal complications, adverse perinatal outcomes, postnatal structural abnormalities, and imprinting disorders add to the complexity and understanding of ASD.

## 10. Evolution of Autistic Spectrum Disorders

One of the striking features of developmental neuropsychiatric disorders is their prevalence in human populations today. This raises the question of what evolutionary forces drive the emergence and sustainability of autism. Here, we theoretically explore the evolution of ASD against the background of epigenetic reprogramming signatures. Humans are a highly social species relative to other mammalian phyla or lineages. Indeed, sociality has been a major driving force in terms of social intelligence, altruistic behavior, and inter- and intra-personal safety. All of these social traits have been conserved and propagated across generations because they have a strong survivorship bias. If ASD first emerged during the evolution of hominin sociality, then under which ecological, cultural, and/or demographic condition(s) did autistic symptoms evolve? Here, we postulate two ecological variables for the likely emergence of ASD: low population density and low fecundity rate.

Low Population Density: If a given population size is relatively small, then it is conceivable that mild or subtle autistic behaviors are less likely to be recognized by the perceptual-cognitive neurons of the human cortex. That is, the introduction of a novel disease into a small, naive population would not greatly disrupt the social networks of the group as disease recognition would be limited. This is in sharp contrast to repeated outbreaks of infectious diseases, for example, where the risk of population crashes (regardless of size) would invariably increase the perceptual–cognitive ability of cortical neurons to recognize infected conspecifics.

Low Fecundity Rate: If a given population has a significant decline in natural fertility, then survival bias for the offspring would increase at the cost of reduced selection pressures. In this context, reproduction and survival variables are intrinsically linked in the life history of genomes as they trade-off tactics with each other in response to uncertainties in ecology [66,67]. Therefore, if the population’s birth rate is relatively low, it would be acceptable to tolerate autistic behaviors. It is possible that, in this case, improved reproductive fitness offsets the costs of having symptomatic kin.

In general, under low population density, disease recognition (i.e., survival) would be repressed, whereas under low fecundity rates, care-giving behavior (i.e., reproduction) would be enhanced. Importantly, this trade-off has relatively low health risks for both the caregiver (i.e., the mother) and group kin. In line with reproduction and survival tactics, survival cancels out the benefits of reproduction in certain ecological variances. In other ecological arrangements, reproductive fitness outweighs survival bias. The end result of this non-adaptive commutation is that ASD has had NEUTRAL evolutionary origins that have been maintained and stabilized across time scales through transgenerational epigenetic inheritance. A NEUTRAL theory of phenotypic evolution involving either the ON or OFF expression of nuclear and mitochondrial genomes according to life history tactics would explain (in part) the emergence of neuropsychiatric ailments such as ASD.

## 11. Epigenetics and Autoimmunity in ASD

There is now substantial evidence implicating a dynamic interplay between gene variants and environmental fluctuations in the development of individual differences in behavior and health. Epigenetic mechanisms that either reduce or increase susceptibility to pathogenic environments have an impact on health outcomes in particular. As discussed earlier, it is likely that ASD first emerged in naive populations with large and complex social formations. Indeed, it is thought that modern humans underwent a dramatic change not only in DNA base sequence or its non-coding regulatory elements, but also in dietary transitions, social kin networks, migration patterns, shifting patterns of fecundity and survival, and growing population densities more than 10,000 years ago [68,69,70]. These large and complex social transformations also brought in increased transmission loads of ancient and newly evolved microorganisms, including viral and bacterial pathogens causing infectious diseases [71,72]. For instance, the transition from foraging for wildlife to domesticating vertebrate species resulted in humans being exposed to zoonotic diseases within the confinement of large and densely populated settlements. This demographic transformation created the landscape ideally suited for socially transmitted diseases to infect fetal tissues, including the nascent central nervous system. If infections prevailed, the secretion of cytokines (e.g., interleukin-1B, IL-1B), T cells (e.g., CD4+, CD8+), and other proinflammatory molecules (e.g., tumor necrosis factor, TNF) by the maternal and fetal immune systems could have led to poor neurological outcomes, including diseases that manifest much later in postnatal life [68,69]. Against this pathogenic background, there is evidence for the presence of maternal autoantibody groups directed against naive proteins of the developing fetus. This observation suggests new pathophysiological hypotheses regarding the molecular origins of autism.

There is growing evidence that dysregulation of the maternal (adaptive) immune system can erode the behavior of pluripotent stem cells that contribute to brain and bilateral body plan development. In fact, the Maternal Antibodies Related to ASD (MAR ASD) subtype of autism is an illustration of immunity gone wrong in pregnant women who unknowingly produce autoantibodies with particular binding properties against whole or fragmented fetal proteins (i.e., antigens). Although these autoantibodies share many common features and can function synergistically to drive disease onset, they appear to differ mechanistically and have unique clinical features. Regardless, eight maternal autoantibodies related to MAR ASD have thus far been identified: CRMP1 and CRMP2, GDA, LDH-A and LDH-B, NSE, STIP1, and YBX1 [72,73]. Overall, two or more of these autoantibodies are required to break immune tolerance and increase the risk of developing symptoms associated with autism (Table 3). Most recently, three fetal antigens targeted by autoantibodies have been recognized: RPL23, GAPDH, and CAMSAP3 [73]. These peptides and proteins are predominantly found on neurons and are therefore likely candidates for the manifestation of full-blown MAR ASD. However, it is not yet clear whether autoantibodies strike neurons before the emergence of symptoms or whether the strike occurs concurrently with the development of disease. Also, it is not clear whether the maternal immunological memory relies on epigenetic remodeling or whether the immunological responsiveness of the fetus to autoantibody groups is epigenetically linked. Regardless, the adaptive and innate responsiveness of both the mother’s and child’s immune systems appear to be governed by complex epigenetic mechanisms [71,72,73,74]. Dysregulation of these epigenetic mechanisms, as well as genomic mutations at imprinted gene clusters, may lead to cognitive disability, deficits in social–emotional responsiveness, repetitive or stereotyped motor behavior, improperly coordinated language communication, and gastrointestinal distress.

## 12. Outlook on Epigenetic Diagnostics and Treatment of ASD

The CDC has been keeping track of ASD prevalence in the US since 1996. The CDC recorded the highest ASD prevalence to date in its most recent data from 2018, indicating 230.0/10,000 children, or 1 in 44 children [75]. While some argue that this is due to improved diagnoses, others argue that there has been an actual increase [76,77,78]. With this work, we are considering the potential that this tendency may be partially explained by environment-induced epigenetic alterations, primarily the hypermethylation of CpG islands in a growing list of gene promoters (e.g., regulator of apoptotic cell death BCL-2, oxytocin receptor OXTR) [79,80]. Risk factors potentially linked to the development of ADS include maternal–fetal stress; indicators of maternal well-being and health conditions during pregnancy (e.g., obesity, diabetes, hypertension); maternal nutritional status, including dietary supplements (e.g., vitamins); and environmental toxicants, as previously reviewed [81,82]. The idea that environmental clues offer some intriguing insights into screening, diagnosis, and care is based on epigenetics. Beginning in the early 2000s, cancer detection and, more specifically, investigations into the treatment of epithelial malignancies have pioneered epigenetic diagnostics and drugs [83,84]. These works cover the entire spectrum of epigenetic modifications, ranging from CpG methylation, histone code and modifications, and various non-coding RNA species (e.g., micro- and long non-coding RNA). This knowledge is currently used to inform diagnostic strategies (e.g., determination of tumor risk), and treatment strategies. Comparable research with an ADS focus is still in its infancy, and while the use of epigenetics in prevention, diagnosis, and therapy shows promise, much more research is needed due to the complexity of embryonic development, unforeseen implications of epigenetic treatments, and the ethical concerns that result. It will be necessary to conduct more solid pre-clinical research followed by large multicenter clinical trials to develop workable epigenetic strategies for the prevention, diagnosis, and treatment of ASD. The fact that ASD is a heterologous disorder also needs to be considered. To completely understand and treat the condition, a multi-pronged strategy involving genetics, epigenetics, neurology, and clinical investigations is required.

## 13. Conclusions

Past experiences can shape current and future behavior through epigenetic mechanisms that can alter gene expression in individual cells, cell ensembles, and whole organisms. These experiences—in particular, chronic adversities and acute stressful events—can lead to clinical consequences such as those diagnosed in ASD. However, ASD is not only the consequence of environmental adversities, but also of discrete functions of wild-type DNA, mutant monoallelic variants, and non-coding regulatory mechanisms that control the transcription, splicing, stability, and translation of nearby or distant genes. In addition, environmental conditions that have shaped human physiology, health, and lifestyle over the course of the last 10,000 to 15,000 years have left an indelible imprint on individual cells that impart either an enhanced or a blunted susceptibility to disease outcome.

## Figures and Tables

**Figure 1 genes-14-01734-f001:**
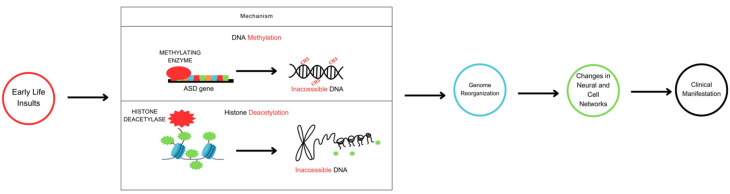
Schematic diagram of epigenetic mechanisms in the development of ASD. Early exposure to stressful adversities or environmental insults leads to epigenetic reprogramming in neurons and peripheral cell networks via DNA methylation and histone deacetylation levels. Such rearrangements cause genomic instability and inappropriate nuclear and cytoplasmic gene expression. In the nascent brain, for example, alterations in gene expression might erroneously guide cell proliferation and differentiation, synapse formation, or neurotransmitter release toward a particular clinical outcome. Similarly, abnormal rearrangements in gene expression outside the pediatric brain could also disrupt cell signaling states to further the clinical course of ASD.

**Figure 2 genes-14-01734-f002:**
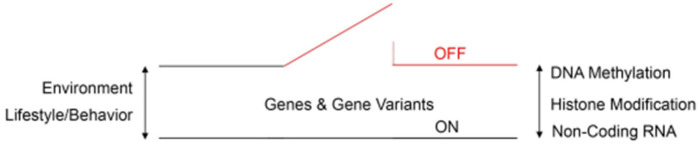
Schematic diagram of an epigenetic switch that shows the complex interactions between epigenetic reprogramming signatures, environmental risk factors, and behavioral risk factors in ASD. Numerous de novo missense mutations across the genome are significantly associated with the risk of autism, according to genome-wide association studies. DNA methylation, histone and non-histone protein modification (acetylation/deacetylation), and non-coding RNA activity define epigenetic reprogramming signatures that can reversibly switch gene activity between ON and OFF states. Epigenetic reprogramming signatures respond to environmental and lifestyle/behavioral clues either toward or away from ASD phenotypes.

**Figure 3 genes-14-01734-f003:**
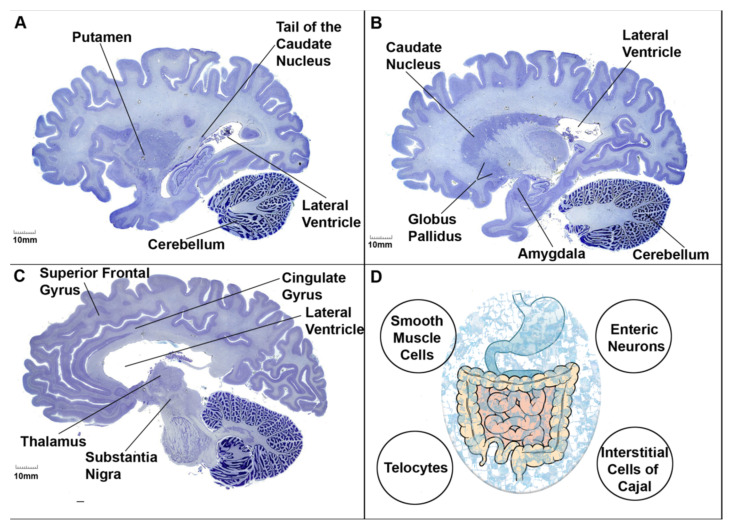
Human brain sections (**A**–**C**) and a schematic diagram of the gastrointestinal tract (**D**). As indicated by the clinical vignette, autistic patients display a number of aberrant behaviors that are linked to anatomical regions of the brain and bilateral body plan. Neurons of the cingulate gyrus (e.g., pyramidal cells), amygdala (e.g., GABAergic cells), caudate putamen (e.g., striatal medium spiny neurons), and cerebellum (e.g., Purkinje cells) establish synaptic connections both within and with other neurons to form distinct circuits underlying behavior. It is now recognized that damage to the synapse or circuit formation in children could lay the foundation for neuropsychiatric disorders. Autistic patients also complain of gastrointestinal motility disorders such as gastroparesis. The rhythmic movement of gastrointestinal content is facilitated by peristalsis, a process mediated by pacemakers (e.g., interstitial cells of Cajal), contractions (e.g., smooth muscle cells), fluid secretion (e.g., enteric neurons), and supporting cell types (e.g., telocytes). The gastrointestinal tract is colonized by trillions of microorganisms (blue marbleized oval), including viral and bacterial lineages that harbor the second gene pool of the bilateral body plan. Gut microbial dysbiosis is thought to indirectly contribute to the pathogenesis of ASD. Human brains: sagittal plane; Nissl stain. For spatial orientation, the lateral ventricles are shown. Images from the Yakovlev–Haleem collection are used courtesy of the National Museum of Health and Medicine, Armed Forces Institute of Pathology. Michigan State University Brain Biodiversity Bank.

**Table 1 genes-14-01734-t001:** Autism-Associated Homeobox Domain Genes with Functional Roles in Brain Development.

Gene ID	Gene Name	GIFtS
*ZEB1*	Zinc Finger E-Box Binding Homeobox 1	56
*PBX1*	PBX Homeobox 1	55
*NKX2-1*	NK2 Homeobox 1	54
*ZEB2*	Zinc Finger E-Box Binding Homeobox 2	54
*OTX2*	Orthodenticle Homeobox 2	53
*NKX2-5*	NK2 Homeobox 5	52
*SATB2*	SATB Homeobox 2	52
*CUX1*	Cut Like Homeobox 1	52
*EMX2*	Empty Spiracles Homeobox 2	51
*ARX*	Aristaless Related Homeobox	50
*SIX3*	SIX Homeobox 3	50
*ADNP*	Activity-Dependent Neuroprotector Homeobox	50
*DLX5*	Distal-Less Homeobox 5	50
*LMX1B*	LIM Homeobox Transcription Factor 1 β	50

Data curated from GeneCards. GIFts: GeneCards Inferred Functionality Score.

**Table 2 genes-14-01734-t002:** Pharmacological Treatment and Management of Autism Spectrum Disorders.

Drug Class	Mechanism of Action	Clinical Applications in Pediatric Disorders	Targeted Behavioral Indices	Individual Side Effects
Risperidone (Atypical Antipsychotic)	D_2_, 5HT_2A_ Receptor Antagonist	ASD	Behavioral Irritability, Stereotyped Behaviors	↑ Appetite, Weight Gain
Aripiprazole (Atypical Antipsychotic)	Partial D_2_ Receptor Antagonist. Partial 5HT_1A_ Receptor Agonist	ASD	Behavioral Irritability	↑ Appetite, Weight Gain
Haloperidol (Typical Antipsychotic)	D_2_ Receptor Antagonist	ASD	Behavioral Hyperactivity	↑ Extrapyramidal Symptoms (Dystonia)
Sertraline (Antidepressant and Antianxiety)	5HT Reuptake Inhibitor	ASD with Fragile X Syndrome	Lexical and Semantic Indices	↑↓ Restlessness and Hyperactivity
Methylphenidate (Psychoactive)	NE and DA Reuptake Inhibitors	ASD with ADHD	Hyperactivity and Attention Deficits	↓ Appetite, Abdominal Pain, Insomnia

Autism Spectrum Disorders can be managed with appropriate pharmacotherapy. Selective dopamine (DA) and serotonin (5HT) based drugs are the mainstay of pharmacological treatment [43,44]. Additional neurotransmitter systems (e.g., norepinephrine (NE) and histamine) are also drug targets. It is not known whether the listed drugs regulate epigenetic mechanisms to counteract autistic symptoms. What is broadly known is that atypical, typical and psychoactive drugs act on DA and 5HT signaling pathways within regions of the human brain (e.g., cortex and basal ganglia) that are behaviorally relevant to the pathophysiology of ASD. Attention Deficit Hyperactivity Disorder (ADHD) and Fragile X Syndrome are debilitating neuropsychiatric conditions commonly diagnosed in pediatric populations. Fragile X Syndrome is a monogenic inherited disease leading to cognitive disability and ASD. Another neuropsychiatric disorder with social deficits and stereotyped behaviors similar to those of autism is Angelman Syndrome (AS). Loss of activity of the maternally inherited *UBE3A* gene causes AS. This particular imprinted gene codes for an E3-ubiquitin ligase, which is critical for synaptogenesis throughout brain development [45]. Altogether, these clinical observations suggest that ADHD, Fragile X Syndrome and AS share several epigenetic and phenotypic features with ASD. Arrows indicate increasing or decreasing medication side effects. In some cases either one can be observed.

**Table 3 genes-14-01734-t003:** Examples of Antigen Combinations Associated with MAR ASD.

Examples of Antigen Combination	% Found in Mothers of Children with ASD	% Found in Mothers of Children without ASD
LDH + YBX1	2%	0%
YBX1 + CRMP2	5%	0.6%
LDH + GDA + STIP1	5%	0.6%
LDH + CRMP1 + STIP1	5%	0%
STIP1 + CRMP1 + GDA	7%	0.6%
LDH + CRMP1 + STIP1 + GDA	2%	0%

## Data Availability

Data sharing not applicable. No new data were created or analyzed in this study. Data sharing is not applicable to this article.

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
