# Peer review of "Conceptualizing Epigenetics and the Environmental Landscape of Autism Spectrum Disorders"

_genes, 2023, doi:10.3390/genes14091734_

Round 1
Reviewer 1 Report
This is a very interesting and well-written paper, but it is less a review (as might be suggested by the title) than an elucidation of the authors' conceptualization of this complex field. This is not necessarily a complaint, as I would agree that the field needs more and better concepts. However, it is not going to be useful to readers who are looking for a review of current research related to epigenetic contributions to the autism spectrum disorders.
For one thing, it is fairly sparsely referenced, and the choice of references is quite individual. In addition, it occasionally makes statements that are highly questionable: eg., most persons working in the field of developmental disorders would not ascribe deterministic gene or injury causes to most cases.
All of that being said, this contribution would seem to be worthwhile, but perhaps should be retitled to reflect the fact that it is predominantly an attempt to conceptualize the role of epigenetics with respect to autism spectrum disorders in an individual fashion.
Author Response
Dear Reviewer. Thank you for your time and expertise! We have implemented the following changes:
1) The title of the manuscript was revised to reflect the conceptualizing nature of this review.
2) Additional references were added in support of a new paragraph at the end tying important concepts together and providing an outlook regarding diagnostic and treatment opportunities of ASD cases underlying epigenetic causation.
3) 'Developmental deviations' were added to broaden causation driving ASD and to clear up the statement deemed questionable.
Thank you - your input is very much appreciated!
Reviewer 2 Report
This is a nice and well-written review manuscript presenting important data regarding the evidence of DNA methylation changes and histone deacetylation abnormalities in individuals with autism spectrum disorders. Figures 1 and 2 present the main data summarized to the general reader. My suggestion to the authors is to present briefly how was the review manuscript produced regarding its methodology (i.e., was it a narrative review?, was it produced after a bibliographic review by a systematic review?). Another suggestion is to change the designation of autism from a "psychiatric condition" to a "neuropsychiatric and neurodevelopmental disorder". Despite not being the aim of the manuscript, my suggestion is to add a brief topic discussing the importance of the pathophysiological mechanisms and changes in DNA methylation and histone deacetylation on therapeutic purposes for autism spectrum disorders.
Author Response
Dear Reviewer. Thank you for your time and expertise. Our manuscript has significantly improved thanks to your recommendations. We have implemented the following suggested changes:
1) A brief review strategy was added to the end of the introduction.
2) We are now referring to ASD as neuropsychiatric and neurodevelopmental disorder
3) An additional paragraph was added highlighting the status of ADS and potential diagnostic and treatment opportunities that arise from epigenetic causation. Such approaches have been developed for certain cancers with encouraging perspectives for other diseases, including ASD.
Your input is appreciated - Thank you!
JL
Reviewer 3 Report
This review is very interesting and provides new suggestions in understanding the pathophysiological mechanisms underlying autism.
The review is well structured and the genomic, epigenetic and immunological approaches proposed and developed are relevant.
Author Response
Dear Reviewer. Thank you for endorsing our manuscript!
We made a few improvements going through the review process and you might enjoy an additional paragraph added towards the end of the manuscript.
Thank you again and best wishes
JL
Reviewer 4 Report
The current draft reviewed the epigenetics of autism spectrum disorder (ASD). Following a thorough review, it appears that the current manuscript has room for improvement.
1. Overall, I’d like the main text, especially the introduction, to be more lay-man friendly. The authors should also appropriately refer relevant literature, for example, “they are thought to nevertheless occur directly via genetic mutations that have disease-driving effects or irreversible brain injury that leads to premature cell death or necrosis…” in Introduction. The authors should use correct scientific terminology. For example, the abstract's precise wording "epigenetic memories" may not effectively express the scientific understanding in this situation. Another example is the term "naive populations," which may be unfamiliar to certain readers. I'd advise the authors to double-check the document for clarity and conciseness. Also, it would be nice if the authors could clarify a scientific word, such as bilateral body plan, when it first appears.
2. Given that the authors touched on the genetic variants linked to ASD, it would be excellent if they could briefly cover the common and rare genetic variants, respectively.
3. Please keep in mind that ASD and Autism are not the same thing. The current paper appears to conflate these two words. For example, the title of each section has the word Autism, while the text within those sections uses the word ASD.
4. When reviewing the prevalence of ASD, I'd like the authors to clearly discuss the exact prevalence of ASD over the last decade as well as the shifting trends.
5. It would be beneficial if the current review included a section summarizing the ASD-associated environmental factors, as well as whether any studies have specifically documented that epigenetic alteration plays a role in the environmental effect on the development of ASD.
6. The major numbers' quality might be improved further. For example, in picture 3, I am unable to read the text annotation.
7. Based on the review, I anticipate a discussion on potential future research directions by the end of this draft.
8. Clarity of Writing:
- “Epigenetics refers to changes in DNA that do not involve modifications to the underlying nucleotide base sequence of a particular gene…” I think it is more common to refer DNA methylation as methylation modification on DNA than changes in DNA.
- “hundreds of gene variants associated with the onset of ASD” Do you mean “genetic variants”?
- Please remember to unify the font size.
Please see my comments above.
Author Response
Dear Reviewer. Thank you for your time and expertise. We think that this manuscript has significantly improved thanks to your advice. The following changes were implemented:
1) To make the text more palatable to non-scientific readers and to improve clarity and conciseness we either replaced or explained questionable terminology as pointed out.
Additional benchmark references were added to a new paragraph toward the end of the manuscript highlighting the status quo of ASD and future perspectives.
2) The focus of this review lies on epigenetic causation. We are introducing a novel homeodomain-based strategy for the identification of genetic variants linked to ASD. In this paragraph, we are referring to established autism candidate genes (references 36-41).
3) We detangled autism from ASD and are now consistently using ASD. We acknowledge that autism falls under ASD.
4, 5, 7) As requested, we developed an additional paragraph toward the end, that ties in together prevalence, associated environmental contributors (also discussed earlier in the manuscript), as well as future diagnostic and treatment opportunities emerging from epigenetic causation. Certain cancers are already diagnosed and treated based on epigenetics which we are using as an example.
6) We have completely redrawn Figure#3. Now the text size is consistent throughout, and, more importantly, legible. Thank you for catching that.
8) We agree and changed 'in DNA' to 'on DNA'
As mentioned, the font size in Figure 3 is now unified. Everything else is going to be unified as part of the editorial process.
Thank you very much - your contributions are appreciated!
JL